# Is experience of the HIV/AIDS epidemic associated with responses to COVID-19? Evidence from the Rural Malawi

Philip Anglewicz[1]*, Sneha Lamba[2], Iliana Kohler[3], James Mwera[4], Andrew Zulu[4], Hans-Peter Kohler[3]

1 Department of Population, Family and Reproductive Health, Johns Hopkins Bloomberg School of Public Health, Baltimore, Maryland, United States of America, 2 Department of Development Economics and Centre for Modern Indian Studies (CeMIS), University of Göttingen, Waldweg Göttingen, Germany, 3 Population Studies Center, University of Pennsylvania, Philadelphia, Pennsylvania, United States of America, 4 Invest in Knowledge Initiative (IKI), Zomba, Malawi

* panglew1@jhu.edu

## Abstract

### Introduction

Starting in late 2019, the coronavirus "SARS-CoV-2", which causes the disease Covid-19, spread rapidly and extensively. Although many have speculated that prior experience with infectious diseases like HIV/AIDS, Ebola, or SARS would better prepare populations in sub-Saharan Africa for COVID-19, this has not been formally tested, primarily due to data limitations.

### Methods

We use longitudinal panel data from the Malawi Longitudinal Study of Families and Health (MLSFH, waves 2006, 2008, and 2020) to examine the association between exposure to the HIV/AIDS epidemic and perceptions of, and behavioral response to, the COVID-19 pandemic. We measured exposure to HIV infection through perceived prevalence of HIV/AIDS in the community, worry about HIV infection, perceived likelihood of HIV infection, and actual HIV status; and the experience of HIV/AIDS-related mortality through self-reports of knowing members of the community and extended family who died from AIDS (measured in 2006 or 2008). Our outcome measures were perceptions of COVID-19 presence in the community, perceptions of individual vulnerability to COVID-19, and prevention strategies to avoid COVID-19 collected through phone-interviews in 2020.

### Results

Based on our data analysis using multivariable regression models, we found that the experience of HIV-related mortality was positively associated with perceptions of COVID-19 prevalence in the community and preventive behaviors for COVID-19. However, perceived vulnerability to HIV-AIDS infection and actual HIV positive status 10-years prior to the COVID-19 pandemic are generally not associated with COVID-19 perceptions and behaviors.

**Data Availability Statement:** All relevant data are within the paper and its Supporting Information files.

**Funding:** This research was supported by Eunice Kennedy Shriver National Institute of Child Health and Human Development, grant R01 HD044228, Principal Investigator: Hans-Peter Kohler; Co-PI, Philip Anglewicz. The funders did not play a role in this research.

**Competing interests:** The authors have declared that no competing interests exist.

## Conclusions

Our results suggest that COVID-19-related behaviors are impacted more by experience of AIDS mortality instead of HIV/AIDS risk perceptions, and that individuals may be correctly viewing HIV/AIDS and COVID-19 transmission as distinct disease processes.

## 1. Introduction

Despite limited resources and overburdened health systems, Sub-Saharan Africa (SSA) has a lower COVID-19 burden than other regions (based on reported cases and deaths) [1]. As of August 2022, the 47 countries of the WHO African region reported nine million cumulative cases and approximately 174,000 cumulative deaths, accounting for close to 1.6% of global cumulative cases and 2.7% of global cumulative deaths [2]. However, with limited COVID-19 testing capacity in much of sub-Saharan Africa [3, 4], the extent of COVID-19 prevalence is more uncertain in Africa than other contexts.

Several explanations have been offered for why SSA has been spared of a higher disease burden and mortality due to COVID-19. In the early stages of the pandemic, SSA governments were praised for their swift action in response to the imminent pandemic, as a number of governments quickly imposed travel bans, mobility restrictions, limitations on public gatherings, and other measures to prevent the spread of COVID-19 [5–8]. Other explanations include a younger population age structure, weather conditions that inhibit the transmissibility of COVID-19, and lower risk of importation of COVID-19 from China and elsewhere [4, 9, 10].

The hypothesis that we investigate in this paper is whether "lessons" learned from previous infectious diseases prepared populations in SSA for COVID-19, and that these prior experiences helped reduce COVID-19 cases and deaths. Some prior studies investigating this hypotheses highlighted the existence of structures and institutions set up for previous epidemics that could be repurposed for COVID-19 [11]; for example, some countries leveraged resources meant for widespread HIV and tuberculosis testing in their response to COVID-19 [12]. Programmatic approaches used for HIV/AIDS, like targeting high risk groups and providing communication and education, were also repurposed for COVID-19 [13]. Community responses to HIV/AIDS risks may also have strengthened local community structures that have subsequently been pivotal in shaping individuals' knowledge about the COVID-19 pandemic and the adaption of preventive health behaviors [14].

In the early stages of the pandemic, prior to vaccines and biomedical treatments, the course of COVID-19 largely depended on non-clinical prevention methods: the extent to which affected populations adopted behaviors that protect from infection. In this context, individual, community, and societal experience with HIV/AIDS and other infectious diseases might affect individual responses to COVID-19, and this aspect might have been particularly important during the early phases of the pandemic [11, 15, 16]. Yet, while several studies have postulated this relationship, the exact mechanisms connecting experience with previous infectious diseases and behaviors related to COVID-19 have not been clearly described. Existing literature focusing on individual experiences is typically conjectural, as there is a dearth of data sources that enable us to test whether individuals who have experienced epidemics such as the HIV, Ebola and SARS in the past are more responsive to the COVID-19 pandemic.

### 1.1. The COVID-19 pandemic in Malawi

The first official case of COVID-19 in Malawi was recorded on April 2, 2020. In the following weeks, sporadic transmission clusters emerged in large cities and additional importations of

COVID-19 cases occurred among migrants returning primarily from South Africa, the largest documented outbreak across Africa countries at the time [17, 18]. The Malawian Government declared COVID-19 as a national disaster as early as March 2020, and banned the gathering of more than 100 people [19]. Malawi joined several other southern African nations in announcing a three-week lockdown on the 18[th] of April 2020. However, the lockdown could not be implemented after it sparked protests and was reversed by civil society groups through a court injunction [14, 20]. The first wave of the pandemic in Malawi coincided with a politically volatile period, with a new government formed in June 2020. Despite a transition in government, Malawi continued with a multi-sectoral pandemic response. In addition to restrictive policies such as international travel ban, school closures at all levels, cancellation of public events, decongesting workplaces and public transport, and mandatory face coverings, the government instituted supportive interventions such as risk communication and community engagement in multiple languages and over a variety of mediums such as national radio, interactive phone texts, community awareness meetings, and distribution of printed materials [14, 21].

In the early days of the COVID-19 pandemic, experts voiced concerns about the potentially huge toll that COVID-19 could have in SSA. In addition to COVID-19 related mortality, lockdowns, closures, and restrictions could have severe consequences for millions of Malawians [14, 22]. Some criticized the swift and restrictive government action in Malawi- and other countries in SSA- noting that measures such as lockdowns and stay at home orders may be ill-suited for low-income countries where people live in close quarters and physical distancing exacts a heavy toll on people's livelihoods [23]. Although the toll of the first wave of the pandemic between March and August 2020 was modest in Malawi, caseloads and death toll of the second and third waves in early 2021 and mid 2021 were more devastating: early in July 2021, the World Health Organization (WHO) and Centers for Disease Control (CDC) designated Malawi to be a high-risk country following an unprecedented rise in cases. There were 48,086 COVID-19 cases and 1,458 deaths from COVID-19 reported in Malawi between January 2020 and 25[th] July 2021 [24]. By May of 2023 this had increased to 88,638 cumulative cases, and 2,686 cumulative deaths reported [2].

## 1.2. The HIV epidemic in Malawi

Among countries in SSA, Malawi has a moderate HIV/AIDS epidemic, with roughly 1 million people living with HIV/AIDS in 2021 in a total population of approximately 18 million [25]. By the peak of the HIV epidemic in 2004, there were at least 74,000 HIV-related deaths occurring per year in Malawi [25]. In the MLSFH study sample aged 15–49, 5.8% (women: 7.0%; men: 3.7%) were HIV infected in 2008 [26]. In the last decade or so, antiretroviral therapy (ART), a highly effective drug treatment that slows the progression of AIDS, became widely available in Malawi [27]. The ART program expanded in Malawi from 2003, when the Malawian Government began to provide ART for free supported by donor funds. The Ministry of Health (MoH) began providing free ART in June 2004 at nine clinics, and by 2008 ART provision has expanded to the MLSFH study regions. By the end of 2010, the number of clinics providing ART had grown to nearly 300 with over 350,000 patients ever initiated on ART [27].

HIV/AIDS infected some and affected everybody in Malawi. Discussions about HIV risk and prevention strategies also became part of everyday life for men and women in Malawi, ranging from graveside condolences to chatting on the bus and interactions in social networks [28–30]. Research using both qualitative and quantitative data collected by the MLSFH showed that Malawians were knowledgeable about the risk factors and well positioned to assess their personal risk of infection through an assessment of their own and spousal behavior [31], and these risk perceptions were often influenced by social interactions [32]. Further, HIV positive

individuals took up many viable prevention techniques including divorcing their partner as risk mitigation strategies and careful selection of new partners [33, 34].

Our analyses use a longitudinal dataset that followed individuals over time from the HIV/ AIDS epidemic through the COVID-19 pandemic. These data come from the Malawi Longitudinal Study of Families and Health (MLSFH), which has been implemented in three sites of rural Malawi since 1998 [26]. Our analyses of the connection between experiences during the HIV/AIDS epidemic and behaviors during the early COVID-19 pandemic are guided by an extension to the Health Belief Model (HBM) that incorporates past experiences of infectious diseases and individual traits to understand how the experience of HIV/AIDS may be associated with individual beliefs about COVID-19, and how these may transition to individual preventive behavior strategies for COVID-19. The HBM then leads us to examine whether exposure and responses to the HIV/AIDS epidemic, such as perceived risk of HIV infection, perceived prevalence of HIV/AIDS, and actual HIV infection status are associated with similar perceptions and behaviors related to COVID-19.

## 2. Theoretical framework

The key question to be investigated in this paper is whether experiences of and responses to the HIV/AIDS epidemic facilitated and affected behavioral responses to the novel COVID-19 pandemic in 2020. In the context of infectious disease, health behavior change models such as the Health Belief Model (HBM) treat risk of infection and the perception of risk as necessary precursors to behavior change [35]. The HBM thus posits that individuals are likely to take action to avoid a health condition if they believe that (1) they are at risk of a potential threat to their health, (2) the condition would have potentially serious consequences, (3) there are behaviors available to them that reduce the susceptibility to or severity of the condition [36]. This serves as the starting point for our conceptual framework.

The HBM does not, however, specifically incorporate prior experience with infectious diseases and how this might affect perceived risk and behaviors regarding current infectious disease environments and epidemics. Individuals who have had experience with a previous pandemic such as HIV/AIDS may be more attuned to preventive measures for a subsequent pandemic at the individual level because the experience affected their "health concern" or makes health issues more salient or relevant [35]. We hypothesize that experience of one infectious disease epidemic such as the HIV/AIDS pandemic in Malawi may affect an individual's perceived vulnerability to another infectious disease such as COVID-19 through several possible channels. These resulting changes in individual perceived vulnerability may or may not lead to individuals undertaking preventive behaviors strategies for the current infectious disease environment, such as wearing face masks or hand washing to prevent COVID-19. This theory of change is displayed in Fig 1.

The experience of the HIV/AIDS epidemic may thus have long-term implications, and when COVID-19 pandemic emerged, may lead to changes in individual beliefs of perceived prevalence of COVID-19 in the community and perceived likelihood of own risk of contracting COVID-19, through three pathways: (1) direct impact, (2) cumulative experience, and (3) mortality impact.

**Pathway 1 (direct impact, measured by actual HIV infection):** HIV infected individuals may be worried about contracting COVID-19 due to their immunocompromised status.

**Pathway 2 (cumulative experience, measured by perceived risk or vulnerability to HIV infection):** Exposure to one infectious disease epidemic affects one's perceptions about infectious disease, and makes individuals more attuned and responsive to other infectious diseases.

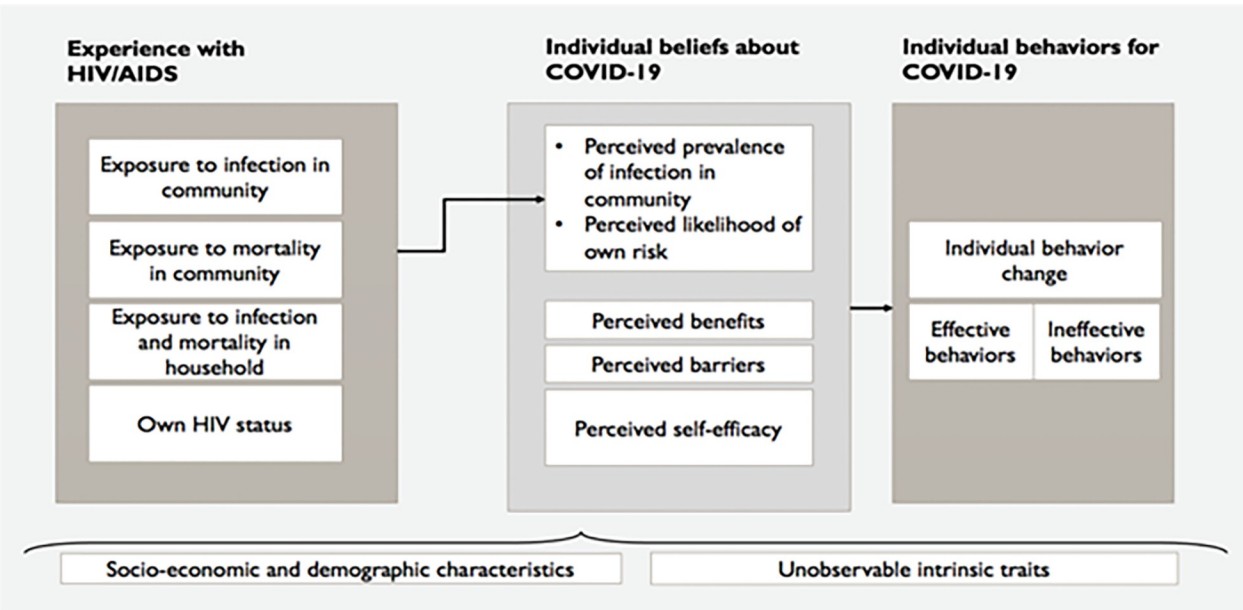

**Fig 1. Conceptual framework: HBM adapted for experience with prior disease.** Note: Adapted by authors from Champion and Skinner (2008).

**Pathway 3 (mortality impact, measured by experience with HIV/AIDS mortality):** Experiencing HIV-related mortality in the household and/or community impacts how individuals respond to infectious diseases in the future.

In addition to these pathways, there may be other reasons why HIV/AIDS-related characteristics are associated with COVID-19 that are not actually due to prior exposure to the HIV/AIDS epidemic. For example, some individuals may be inherently optimistic or pessimistic about the prevalence and their vulnerability from all diseases or threats and this may show up as a positive association between perceptions regarding HIV/AIDS and COVID-19. But this would represent a connection due to individual characteristics and not due to the HIV/AIDS epidemic.

Finally, there may not be any connection between HIV/AIDS and COVID-19. As described above, the prevalence and risk of mortality differ considerably between COVID-19 and HIV/AIDS, with AIDS responsible for 74,000 deaths per year at its peak, compared to 2,686 COVID-19 deaths through May 2023. The mode of transmission also differs between COVID-19 and HIV/AIDS, with the former spread primarily when an infected person breathes out droplets and very small particles that contain the virus to others. Since Malawians are generally well-informed about HIV epidemiology, individuals may have quickly understood the different modes of transmission between HIV/AIDS and COVID-19 and may thus have perceived both as distinct epidemics with separate behavioral responses and risk reduction strategies. Therefore, rural Malawians may believe that exposure to one infectious disease, HIV during the 1990s and 2000s, may not affect individual perceptions and behaviors for another infectious disease.

## 3. Methods

### 3.1 Data

The MLSFH is well-suited to examine this question as it follows a cohort through both the HIV and the COVID-19 epidemics. The MLSFH was initiated in 1998 as a longitudinal

couples' survey, targeting a population-based sample of approximately 1,500 ever-married women and 1,000 of their husbands in three rural sites of Malawi [26]. Since the baseline survey in 1998, the MLSFH has followed up with this cohort in 2001, 2004, 2006, 2008, 2010, 2019, and 2020. The initial focus of the MLSFH survey was on family planning, which shifted more to HIV/AIDS by 2004. MLSFH also offered HIV testing to participants, first in 2004 and with follow-up HIV testing in 2006 and 2008. The MLSFH had very low refusal rates for surveys and HIV testing: over 90% of respondents agreed to be tested for HIV and nearly all tested respondents received their test results in 2006 and 2008. Further description of the MLSFH data, sampling, and HIV testing are presented in Kohler et al., 2015 [26].

In 2020, the MLSFH carried out a phone survey of the COVID-19 pandemic among the original MLSFH study cohort between June 2nd and August 26th, 2020. The 2020 phone survey collected a comprehensive range of information on COVID-19 related topics ranging from knowledge about transmission pathways and behavioral responses to reduce infections, experience of current and past COVID-19 infection symptoms among respondents and members of their households, impact of the pandemic on economic and social well-being, trust in institutions, village-level responses to the pandemic, to subjective well-being and mental health during the pandemic [14].

In this analysis, we used data from the 2006 and 2008 rounds of MLSFH (rounds 4 and 5) and 2020 phone survey, for both men and women. We included respondents who completed either 2006 or 2008 and the 2020 phone interview, which yielded a sample size of 1,394. For respondents who were interviewed in both 2006 and 2008, we used the 2006 measure because it was closer in time to the peak of the HIV/AIDS epidemic in Malawi (results of analysis of 2008 measures for the same respondents yielded the same substantive results); 80.8% of responses are from 2006, 19.2% from 2008. These data were selected strategically, as they include the key HIV/AIDS measure of interest (such as actual HIV test results, with nearly all respondents receiving results), and were collected prior to when ART was widely available in Malawi. We did not include the 2010 round of the MLSFH because it was missing the measure of actual HIV status; however, analysis of other measures yielded results that were not substantively different from those shown here.

The data that the MLSFH collected for this study was approved by the IRB Board at the University of Pennsylvania (IRB Protocols 815016 and 826828), and in Malawi, by the Ethics Committee of the College of Medicine, Malawi (COMREC, Protocols P01/12/1165 and P.04/17/2160) and the National Health Sciences Research Committee (NHSRC, Protocol 19/01/2214).

## 3.2 Measures

Our independent variables capture HIV infection, exposure to and the perceived risk of HIV infection, and HIV/AIDS-related mortality. We also distinguished between varying degrees of exposure to and experience of HIV infection and HIV related mortality in the community and household through our measures. Specifically, we tested the "direct impact" pathway by measuring the association between HIV tatus (measured through HIV testing) and responses to COVID-19. We measured mortality impact through the experience of HIV/AIDS mortality in the household and community. We describe these measures further below.

Exposure to HIV infection and perceived risk of HIV infection was measured through self-reports of: (i) perception of the current prevalence of HIV in the community, measured by asking respondents how many individuals out of 10 they think are currently infected with HIV/AIDS; (ii) worry about being infected with HIV, measured with a three category response of "worried a lot" "worried a little" and "not worried at all"; (iii) perceived likelihood of currently

being infected with HIV, measured with a four category outcome of "none", "low", "medium", and "high.

We measured exposure to AIDS mortality through reports of: (i) number of people known to the respondent who died from AIDS ever; (ii) number of people known to the respondent who died from AIDS in the last 12 months; and (iii) any mortality in the household in the last two years of the survey. For each of the mortality-related variables we created categories that capture varying intensities of experiences of mortality in the community and within the household. For example, for the measure of number of people who died from AIDS in the last 12 months known to the respondent we used the categories: (i) knows no one who died of AIDS in the last 12 months, (ii) knows 1–2 individuals who died of AIDS in the last 12 months, (iii) knows 3–5 individuals who died of AIDS in the last 12 months, and (iv) knows 6 or more individuals who died of AIDS in the last 12 months. These categories are shown for each of our exposure to AIDS mortality measures in Table 1.

The key dependent variables for this paper are related to perceptions of and the response to COVID-19. These measures include: (i) perceived prevalence of COVID-19 infection in the community (currently and in the future); (ii) perceived risk of own likelihood of infection (current and in the future); and (iii) behaviors adopted to avoid infection with COVID-19 (whether effective or not). Respondents were asked if they took up the following preventive actions: (i) used herbs; (ii) washed hands; (iii) avoided close contact; (iv) stayed home; (v) used vaccinations; (vi) used traditional practices; (vii) used a face cover; (viii) avoided shaking hands; (ix) sneezed into their elbows; and (x) prayed. We note that most of these are generally effective strategies to avoid COVID-19 infection, but others, like prayer and herbs, are not considered effective. Households were also asked if they had face masks at home and whether they were socially distancing. We created both dichotomous measures of whether the individuals report undertaking each of these individual behaviors and composite preventive behavior measures for COVID-19: (i) total number of preventive actions for avoiding COVID-19 and (ii) total number of effective preventive actions.

Finally, we also included sociodemographic controls in our regression models. In our models we adjusted for MLSFH region (Northern, Southern, Central), wealth quintile (created using principal components analysis of household asset ownership), sex, age, level of education (none, primary, secondary or higher), marital status (currently married or partnered compared to widowed and divorced/separated), number of living children, and number of household members. The few measures with missing items were dropped from the analysis. The MLSFH survey instruments from 2006 and 2008 can be found online at https://web.sas.upenn. edu/malawiresearch/documentation/questionnaires/, and the MLSFH COVID-19 survey instrument is available as S1 File for this manuscript. A detailed description of the MLSFH data collection approach can be found in [26].

## 3.3 Analytic methods

We begin by presenting tabulations of sociodemographic characteristics, COVID-19-related outcomes, and HIV/AIDS characteristics using the MLSFH data from 2008, 2010, and the 2020 COVID-19 survey. For our main analysis, we conducted multiple regression models in which the dependent variables are the measures of COVID-19 listed above, and independent variables of central interest are the HIV/AIDS measures (we also include sociodemographic controls of sex, age, number of living children, number of household members, region of residence, level of education, and wealth quintile).

Because the COVID-19 measures differ in their response structure, we use linear probability regression, log-linear models and logistic models, depending on the nature of the

**Table 1. Sociodemographic characteristics, COVID-19-related outcomes, and HIV/AIDS characteristics among Malawi Longitudinal Study of Families and Health (MLSFH) respondents.**

| Sociodemographic Characteristics (Measured in 2006 or 2008*) | |
|---|---|
| Female | 57.7% |
| Mean respondent age | 37.0 |
| Mean number of household members | 10.8 |
| Mean number of living children | 4.4 |
| *Wealth Quintiles* | |
| Wealth quintile 1 | 14.3% |
| Wealth quintile 2 | 18.0% |
| Wealth quintile 3 | 19.1% |
| Wealth quintile 4 | 22.2% |
| Wealth quintile 5 | 26.4% |
| *Region of Residence* | |
| Central | 36.7% |
| South | 26.2% |
| North | 37.1% |
| *Education Levels* | |
| No education | 19.4% |
| Primary level education | 68.1% |
| Secondary or higher-level education | 12.4% |
| *Marital Status* | |
| Never married | 1.3% |
| Married/living together | 91.7% |
| Divorced, separated, or widowed | 7.0% |
| **COVID-19 Measures (Dependent Variables, Measured in 2020)** | |
| Household has facemasks | 63.1% |
| Is Socially distancing | 69.5% |
| Using herbs | 3.8% |
| Washing hands | 93.4% |
| Avoid close contact | 91.5% |
| Staying home | 75.4% |
| Vaccination | 2.3% |
| Traditional practices | 3.4% |
| Face cover | 60.1% |
| Avoiding shaking hands | 91.4% |
| Coughing/sneezing into elbow | 84.5% |
| Prayer | 67.1% |
| Other | 3.5% |
| Mean # of preventive actions for COVID-19 | 6.4 |
| Mean # of effective preventive actions for COVID-19 | 6.3 |
| Mean # of ineffective preventive actions for COVID-19 | 0.1 |
| **HIV-related measures (Independent Variables, Measured in 2006 or 2008*)** | |
| *HIV Status* | |
| HIV uninfected | 89.4% |
| HIV infected | 3.8% |
| Refused HIV testing | 6.9% |
| *HIV perceptions* | |
| Perception of the prevalence of HIV in community currently | 30.0% |

(*Continued*)

**Table 1.** (Continued)

| Sociodemographic Characteristics (Measured in 2006 or 2008*) | |
| --- | --- |
| Perception of the prevalence of HIV in community in future | 55.0% |
| Worried that you might catch HIV: High | 16.9% |
| Perception: Likelihood of having HIV is medium or high | 9.2% |
| *Mortality in the household in the last 2 years* | |
| Experienced no mortality in household | 12.2% |
| 1–2 individuals died in household | 38.9% |
| 3–5 individuals died in household | 37.6% |
| More than 6 individuals died in household | 11.2% |
| *Knows anyone who died of AIDS ever* | |
| Knows no one who died of AIDS ever | 2.6% |
| Knows 1–2 who died of AIDS ever | 9.5% |
| Knows 3–5 who died of AIDS ever | 31.8% |
| Knows 6–10 who died of AIDS ever | 27.7% |
| Knows 11 or more who died of AIDS ever | 28.4% |
| *Knows anyone who died of AIDS in the last 12 months* | |
| Knows no one who died of AIDS in last 12mo | 23.0% |
| Knows 1–2 who died of AIDS in last 12mo | 46.2% |
| Knows 3–5 who died of AIDS in last 12mo | 25.1% |
| Knows 6 or more who died of AIDS in last 12mo | 5.1% |
| **N** | **1394** |

Notes: *80.8% of responses came from the 2006 MLSFH survey.

dependent variable: for example, our measure of the total number of preventive actions for COVID-19 is right skewed, with most respondents reporting undertaking between 6–8 of 10 possible preventive actions, so we used a log-linear model for these measures. We also included a dummy measure of MLSFH survey wave to account for differences in observations measured in 2006 and 2008. Because relationships may differ by gender, we first present results for all adults, followed by separately for men and women. All analyses were conducted using Stata version 15 (Stata Corp LP, College Station, Texas).

## 4. Results

We present summary statistics for key dependent and independent variables, as well as sociodemographic controls in Table 1. The majority of respondents were female (57.7%), over 90% were currently married, and about 80% had at least primary school education. The Northern region was the most common place of residence (37.1%), with the fewest living in the South (26.2%).

In the 2020 COVID-19 survey, the majority of respondents were aware of COVID-19 disease symptoms, transmission pathways, appropriate behavioral responses, and widespread compliance with social distancing and other preventive behaviors. Individuals perceived almost double the prevalence of COVID-19 and likelihood of contracting COVID-19 in future, compared to the current prevalence and likelihood of being infected with COVID-19. We observed high percentages of effective preventive actions for COVID-19: 93.4% of individuals in our sample reported washing their hands, 91.5% reported avoiding close contact, and 91.4% reported avoiding shaking hands but lower rates are reported for wearing face coverings (60.1%), owning face masks (63.1%), and staying at home (75.4%). We observe comparatively

low percentages of ineffective preventive actions for COVID-19: only 3.8% of the sample reported using herbs, 3.4% of the sample reported using traditional practices, but 67.1% of the sample reported praying as a preventive measure for COVID-19. Much of this information is similar to other studies on COVID-19 knowledge and behaviors in Malawi [17].

Regarding HIV/AIDS-related measures, individuals perceived a higher prevalence of HIV in the future than their perceived prevalence at the time survey data was collected. More than 10% of respondents thought there was a medium or high chance that they were currently infected with HIV, compared to only 5% who were actually infected according to biomarker testing. HIV/AIDS-related mortality was common: more than half of the sample for this study experienced mortality in the household; 12.2% of the sample reported that they had experienced no mortality in their household from any causes in the last two years. We also find that almost 98% of the sample knew at least one person who had died from AIDS ever, and 77.0% of the sample knew at least one person who had died from AIDS in the last 12 months.

Turning to our multivariable regression results, in our tables, the first set of columns, (1)-(4), show the outcome variables that capture perceptions of the prevalence and vulnerability to COVID-19, and columns (5)-(6) show outcome variables that capture preventive behaviors for COVID-19. We conducted linear regression models for outcomes shown in columns (1)-(4) and log linear models for outcomes in columns (5) and (6).

Table 2 shows results for the relationship between individual-level perceptions of the prevalence of HIV in the community and COVID-19-related outcomes. Columns 1 and 2 in Table 2 shows that there is a positive association between perceived prevalence of HIV in the

**Table 2. Multiple regression results for the association between perception of _current_ prevalence of HIV in community in 2006 or 2008 and COVID-19 perceptions and responses in 2020.**

|  | linear | linear | linear | linear | log-linear | log-linear |
|---|---|---|---|---|---|---|
| **Regression type** | (1) | (2) | (3) | (4) | (5) | (6) |
| | *Dependent Variables* | | | | | |
| | **Perception of the prevalence of COVID-19 in community (currently)** | **Perception of the prevalence of COVID-19 in community (future)** | **Likelihood of current COVID-19 infection** | **Likelihood of future COVID-19 infection** | **ln(Total no. of preventive actions for COVID-19)** | **ln(Total no. of effective preventive actions for COVID-19)** |
| **HIV/AIDS in 2006/2008** | | | | | | |
| | All Adults (N = 1394) | | | | | |
| Perception of the prevalence of HIV in community currently | 0.059** | 0.119*** | 0.016 | 0.079 | 0.004 | 0.004 |
| | Men (N = 606) | | | | | |
| Perception of the prevalence of HIV in community currently | 0.052 | 0.021 | 0.053 | 0.088 | -0.012 | -0.011 |
| | Women (N = 788) | | | | | |
| Perception of the prevalence of HIV in community currently | 0.070** | 0.179*** | -0.025 | 0.073 | 0.014 | 0.014 |

Notes: regression models adjust for sex (in models for all adults), age, number of living children, number of household members, region of residence, level of education, and wealth quintile. The HIV/AIDS exposure measure is statistically significant at

\*\*\* p<0.01

\*\* p<0.05

\* p<0.10.

**Table 3. Multiple regression results for the association between perception of _future_ prevalence of HIV in community in 2006 or 2008 and COVID-19 perceptions and responses in 2020.**

|  | linear | linear | linear | linear | log-linear | log-linear |
|---|---|---|---|---|---|---|
| **Regression type** | (1) | (2) | (3) | (4) | (5) | (6) |
|  | *Dependent Variables* | | | | | |
|  | Perception of the prevalence of COVID-19 in community (currently) | Perception of the prevalence of COVID-19 in community (future) | Likelihood of current COVID-19 infection | Likelihood of future COVID-19 infection | ln(Total no. of preventive actions for COVID-19) | ln(Total no. of effective preventive actions for COVID-19) |
| **HIV/AIDS in 2006/ 2008** | | | | | | |
|  | All adults (N = 1394) | | | | | |
| Perception of the prevalence of HIV in community in future | 0.037* | 0.054* | 0.026 | 0.066 | -0.001 | -0.001 |
|  | Men (N = 606) | | | | | |
| Perception of the prevalence of HIV in community in future | 0.051* | 0.037 | 0.038 | 0.095 | -0.004 | -0.004 |
|  | Women (N = 788) | | | | | |
| Perception of the prevalence of HIV in community in future | 0.031 | 0.066* | 0.011 | 0.045 | 0.002 | 0.002 |

Notes: regression models adjust for sex (in models for all adults), age, number of living children, number of household members, region of residence, level of education, and wealth quintile. The HIV/AIDS exposure measure is statistically significant at

\*\*\* p<0.01

\*\* p<0.05

\* p<0.10.

community currently and perceived prevalence of COVID-19 in the community (current and future). Individuals who perceived a one-unit higher prevalence of HIV in the community in 2006/08, perceived a 0.06 unit higher prevalence of COVID-19 in the community currently (p<0.05) and 0.12 unit higher prevalence of COVID-19 in the community in future (p<0.05). In sub-group analysis, we find that this result is statistically significant (at p<0.05 and p<0.01) for women but not for men. There is no statistically significant association between perceived prevalence of HIV and effective behaviors for preventing COVID-19 (in columns 5 and 6 of Table 2). We see similar positive association between perceived prevalence of HIV in the community in future and perceived prevalence of COVID-19 in the community currently, among all individuals and men (column 1 of Table 3), but this is statistically significant only at the p<0.10 level.

Turning to the relationship between perceptions of HIV risk and COVID-19, Table 4 shows that women who were highly worried about being infected with HIV perceived a 0.442 greater odds of future COVID-19 infection, compared to women who were not worried or hardly worried about being infected with HIV (p<0.05). As shown in Table 5, we find similarly find a positive association between perceiving a medium or high likelihood of HIV infection and the likelihood of future COVID-19 infection (for all adults and women), but this is statistically significant only at the p<0.10 level. Besides these results, there are no statistically significant associations between perceptions of HIV risk and COVID-19-related behaviors or perceptions.

**Table 4. Multiple regression results for the association between worry about being infected with HIV in 2006 or 2008 and COVID-19 perceptions and responses in 2020, MLSFH data.**

| | linear | linear | linear | linear | log-linear | log-linear |
|---|---|---|---|---|---|---|
| **Regression type** | (1) | (2) | (3) | (4) | (5) | (6) |
| | *Dependent Variables* | | | | | |
| | Perception of the prevalence of COVID-19 in community (currently) | Perception of the prevalence of COVID-19 in community (future) | Likelihood of current COVID-19 infection | Likelihood of future COVID-19 infection | ln(Total no. of preventive actions for COVID-19) | ln(Total no. of effective preventive actions for COVID-19) |
| **HIV/AIDS in 2006/2008** | | | | | | |
| All adults (N = 1394) | | | | | | |
| Worried that you might catch HIV: High | -0.045 | -0.254 | 0.094 | 0.442** | -0.022 | -0.024 |
| Men (N = 606) | | | | | | |
| Worried that you might catch HIV: High | -0.115 | -0.344 | 0.069 | 0.568 | 0.030 | 0.026 |
| Women (N = 788) | | | | | | |
| Worried that you might catch HIV: High | 0.004 | -0.210 | 0.124 | 0.429 | -0.032 | -0.034 |

Notes: regression models adjust for sex (in models for all adults), age, number of living children, number of household members, region of residence, level of education, and wealth quintile. The HIV/AIDS exposure measure is statistically significant at

*** p<0.01

** p<0.05

* p<0.10.

In Tables 6–8 we show the relationship between experience of mortality (general and HIV/AIDS-specific) and COVID-19 outcomes, starting with mortality in the household. Table 6 shows that individuals who experienced 3–5 individuals dying from any cause in the past two years in their household were more likely to perceive a higher prevalence of COVID-19 in their community currently, compared to individuals who experienced no mortality in their household in the past two years. We find different patterns for men and women; for the former, experience of 1–2, 3–5, and 6+ family members dying in the past two years is associated with lower prevalence of COVID-19 in the future. But among women, we find that experience of 1–2 and 3–5 family member dying in the past two years is associated with higher perceptions of current COVID-19 prevalence. We do not find any statistically significant associations between family mortality and likelihood of COVID-19 infection or COVID-19 preventive behaviors adopted.

We find more associations between mortality and COVID-19 for AIDS-specific deaths. As shown in Table 7, we find that knowing 1–2, 3–5, 6–10, and 11+ individuals who died of AIDS is associated with higher prevalence of COVID-19 currently and in the future, and the likelihood of both current and future COVID-19 infection. These associations are found for both men and women. We find similar associations for AIDS mortality in the past 12 months. Women who knew 1–2, 3–5, or 6+ who died of AIDS in the past 12 months predicted a significantly higher prevalence of COVID-19 in the future. We also find that experience of AIDS mortality in the past 12 months is associated with COVID-19 behaviors in the future for women: women who knew 1–2 and 3–5 who died of AIDS in the past 12 months adopted

**Table 5. Multiple regression results for the association between own likelihood of HIV infection in 2006 or 2008 and COVID-19 perceptions and responses in 2020, MLSFH data.**

| | linear | linear | linear | linear | log-linear | log-linear |
|---|---|---|---|---|---|---|
| **Regression type** | (1) | (2) | (3) | (4) | (5) | (6) |
| | *Dependent Variables* | | | | | |
| | Perception of the prevalence of COVID-19 in community (currently) | Perception of the prevalence of COVID-19 in community (future) | Likelihood of current COVID-19 infection | Likelihood of future COVID-19 infection | ln(Total no. of preventive actions for COVID-19) | ln(Total no. of effective preventive actions for COVID-19) |
| **HIV/AIDS in 2006/2008** | | | | | | |
| All Adults (N = 1394) | | | | | | |
| Perception: Own likelihood of having HIV is medium or high | -0.067 | -0.230 | -0.105 | 0.470* | 0.001 | 0.003 |
| Men (N = 606) | | | | | | |
| Perception: Own likelihood of having HIV is medium or high | -0.154 | -0.282 | -0.347 | 0.289 | 0.062 | 0.064 |
| Women (N = 788) | | | | | | |
| Perception: Own likelihood of having HIV is medium or high | -0.027 | -0.236 | -0.008 | 0.608* | -0.026 | -0.023 |

Notes: regression models adjust for sex (in models for all adults), age, number of living children, number of household members, region of residence, level of education, and wealth quintile. The HIV/AIDS exposure measure is statistically significant at

*** p<0.01

** p<0.05

* p<0.10.

more COVID-19 protective behaviors (both overall and effective) than women who didn't know anyone who died of AIDS in the past 12 months.

Table 9 below shows the relationship between actual HIV positive status and perceptions of the prevalence and vulnerability to COVID-19 in columns (1)-(4) and the relationship between individual's own HIV positive status and preventive behaviors for COVID-19 in columns (5)-(6). We don't find any statistically significant associations between HIV positive status and COVID-19 related perceptions and behaviors.

## 5. Discussion

Although many have suggested that the experience with the HIV/AIDS epidemic would prepare countries and individuals for COVID-19 [11, 15, 16] this connection has not been described in detail or tested empirically- largely due to data limitations. This paper is the first to directly examine the relationship between individual-level exposure to the HIV/AIDS epidemic and responses to the COVID-19 pandemic. To do so, we use longitudinal panel data from rural Malawi that was collected between 2006 and 2019.

We believe that this research has two main contributions. The first is the theoretical framework: previous descriptions of the connection between exposure to the HIV/AIDS epidemic and COVID-19 pandemic have been vague, and we provide more definition of potential

**Table 6. Multiple regression results for the association between all-cause mortality within household in 2006 or 2008 and COVID-19 related perceptions and responses in 2020, MLSFH data.**

| | linear | linear | linear | linear | log-linear | log-linear |
|---|---|---|---|---|---|---|
| **Regression type** | (1) | (2) | (3) | (4) | (5) | (6) |
| | *Dependent Variables* | | | | | |
| | Perception of the prevalence of COVID-19 in community (currently) | Perception of the prevalence of COVID-19 in community (future) | Likelihood of current COVID-19 infection | Likelihood of future COVID-19 infection | ln(Total no. of preventive actions for COVID-19) | ln(Total no. of effective preventive actions for COVID-19) |
| **HIV/AIDS in 2006/2008** | | | | | | |
| | All Adults (N = 1394) | | | | | |
| 1–2 individuals died in family | 0.152 | -0.011 | -0.115 | -0.010 | 0.021 | 0.025 |
| 3–5 individuals died in family | 0.306* | 0.106 | 0.258 | 0.116 | 0.012 | 0.011 |
| More than 6 individuals died in family | 0.218 | -0.324 | 0.339 | -0.038 | 0.015 | 0.014 |
| | Men (N = 606) | | | | | |
| 1–2 individuals died in family | -0.232 | -1.212*** | -0.081 | 0.016 | 0.007 | 0.004 |
| 3–5 individuals died in family | -0.047 | -0.882** | 0.289 | 0.199 | 0.096 | 0.081 |
| More than 6 individuals died in family | -0.199 | -1.091* | 0.516 | 0.056 | 0.099 | 0.088 |
| | Women (N = 788) | | | | | |
| 1–2 individuals died in family | 0.355* | 0.692** | -0.186 | -0.019 | 0.010 | 0.019 |
| 3–5 individuals died in family | 0.444** | 0.587 | 0.194 | 0.006 | -0.065 | -0.057 |
| More than 6 individuals died in family | 0.331 | -0.151 | 0.177 | -0.171 | -0.063 | -0.057 |

Notes: regression models adjust for sex (in models for all adults), age, number of living children, number of household members, region of residence, level of education, and wealth quintile. The reference category in these regressions is no one died in the family in the past two years. The HIV/AIDS exposure measure is statistically significant at

*** p<0.01

** p<0.05

* p<0.10.

pathways through which exposure to HIV/AIDS may impact perceptions of and responses to COVID-19. Using the HBM as a starting point, we create a theoretical framework that explains how and why experience with the HIV/AIDS epidemic could influence behaviors and perceptions related to COVID-19. In this framework we propose three pathways: direct impact, cumulative experience, and mortality impact, while also acknowledging that there may not be a connection between HIV/AIDS and COVID-19 at all. Second, we use data from the MLSFH to empirically test this connection. The MLSFH is well-suited for this analysis, since it was initiated in 1998 and followed individuals through the HIV-AIDS epidemic in the early 2000s and the COVID-19 pandemic in 2020, and also contains appropriate measures on exposure to both HIV/AIDS and COVID-19. We measured exposure to the HIV/AIDS epidemic through

**Table 7. Multiple regression results for the association between AIDS related mortality (ever) in community in 2006 or 2008 and COVID-19 related perceptions and responses in 2020, MLSFH data.**

| | linear | linear | linear | linear | log-linear | log-linear |
|---|---|---|---|---|---|---|
| **Regression type** | (1) | (2) | (3) | (4) | (5) | (6) |
| | | | *Dependent Variables* | | | |
| | Perception of the prevalence of COVID-19 in community (currently) | Perception of the prevalence of COVID-19 in community (future) | Likelihood of current COVID-19 infection | Likelihood of future COVID-19 infection | ln(Total no. of preventive actions for COVID-19) | ln(Total no. of effective preventive actions for COVID-19) |
| **HIV/AIDS in 2006/2008** | | | | | | |
| | | | All Adults (N = 1394) | | | |
| Knows 1–2 who died of AIDS ever | 0.494* | 0.385 | 0.566 | 1.202** | 0.030 | 0.026 |
| Knows 3–5 who died of AIDS ever | 0.675*** | 0.893** | 1.090*** | 1.574*** | 0.032 | 0.023 |
| Knows 6–10 who died of AIDS ever | 0.546** | 0.855* | 0.910** | 1.577*** | 0.056 | 0.046 |
| Knows 11+ who died of AIDS ever | 0.620** | 1.027** | 0.952** | 1.451*** | 0.057 | 0.051 |
| | | | Men (N = 606) | | | |
| Knows 1–2 who died of AIDS ever | 0.246 | 0.452 | 0.193 | 1.391* | 0.037 | 0.024 |
| Knows 3–5 who died of AIDS ever | 0.789** | 0.954 | 0.804 | 1.927*** | 0.013 | 0.007 |
| Knows 6–10 who died of AIDS ever | 0.681* | 0.912 | 0.540 | 1.753** | -0.014 | -0.023 |
| Knows 11+ who died of AIDS ever | 0.744** | 1.098* | 0.620 | 1.737** | 0.063 | 0.062 |
| | | | Women (N = 788) | | | |
| Knows 1–2 who died of AIDS ever | 0.785* | 0.331 | 0.689 | 0.914 | 0.026 | 0.027 |
| Knows 3–5 who died of AIDS ever | 0.725* | 0.949 | 1.200** | 1.216* | 0.066 | 0.054 |
| Knows 6–10 who died of AIDS ever | 0.560 | 0.888 | 1.070* | 1.336* | 0.117 | 0.104 |
| Knows 11+ who died of AIDS ever | 0.664* | 1.086* | 1.112** | 1.138 | 0.078 | 0.065 |

Notes: regression models adjust for sex (in models for all adults), age, number of living children, number of household members, region of residence, level of education, and wealth quintile. The reference category in these regressions is knows no one who dies of AIDS. The HIV/AIDS exposure measure is statistically significant at

*** $p < 0.01$

** $p < 0.05$

* $p < 0.10$.

individual's perceived prevalence of HIV infection in the community and their perceived vulnerability to HIV (cumulative experience), their personal experience of HIV-related morality (mortality impact), and their actual HIV status (direct impact).

Overall, our results show several consistent themes. First, we find that perceptions of prevalence of infectious diseases are linked: perceived prevalence of HIV is associated with perceived prevalence of COVID, both currently and in the future. But perceived prevalence was generally not associated with the adoption of behaviors to avoid COVID-19 infection, or the perceived likelihood of COVID-19 infection. We also find that the experience of AIDS mortality is

**Table 8. Multiple regression results for the association between AIDS related mortality (12mo) in community in 2006 or 2008 and COVID-19 related perceptions and responses in 2020, MLSFH data.**

| | linear | linear | linear | linear | log-linear | log-linear |
|---|---|---|---|---|---|---|
| **Regression type** | **(1)** | **(2)** | **(3)** | **(4)** | **(5)** | **(6)** |
| | | | *Dependent Variables* | | | |
| | **Perception of the prevalence of COVID-19 in community (currently)** | **Perception of the prevalence of COVID-19 in community (future)** | **Likelihood of current COVID-19 infection** | **Likelihood of future COVID-19 infection** | **ln(Total no. of preventive actions for COVID-19)** | **ln(Total no. of effective preventive actions for COVID-19)** |
| **HIV/AIDS in 2006/2008** | | | | | | |
| | | | *All adults (N = 1394)* | | | |
| Knows 1–2 who died of AIDS in last 12mo | 0.162 | 0.439** | 0.248 | 0.403* | 0.020 | 0.017 |
| Knows 3–5 who died of AIDS in last 12mo | 0.030 | 0.269 | 0.112 | 0.111 | 0.071** | 0.066** |
| Knows 6 or more who died of AIDS in last 12mo | 0.079 | 0.379 | 0.133 | 0.512 | 0.040 | 0.027 |
| | | | *Men (N = 606)* | | | |
| Knows 1–2 who died of AIDS in last 12mo | 0.213 | 0.336 | 0.554** | 0.617* | -0.041 | -0.036 |
| Knows 3–5 who died of AIDS in last 12mo | 0.022 | -0.000 | 0.444 | 0.431 | 0.005 | 0.009 |
| Knows 6 or more who died of AIDS in last 12mo | -0.182 | -0.427 | 0.082 | -0.326 | -0.105 | -0.093 |
| | | | *Women (N = 788)* | | | |
| Knows 1–2 who died of AIDS in last 12mo | 0.155 | 0.570** | -0.005 | 0.233 | 0.078** | 0.069* |
| Knows 3–5 who died of AIDS in last 12mo | 0.120 | 0.590* | -0.207 | -0.173 | 0.137*** | 0.123*** |
| Knows 6 or more who died of AIDS in last 12mo | 0.308 | 0.943** | 0.057 | 0.874* | 0.142** | 0.110 |

Notes: regression models adjust for sex (in models for all adults), age, number of living children, number of household members, region of residence, level of education, and wealth quintile. The reference category in these regressions is knows no one who dies of AIDS. The HIV/AIDS exposure measure is statistically significant at

*** $p < 0.01$

** $p < 0.05$

* $p < 0.10$.

associated with protecting from COVID-19 infection, but the exact relationship differs for men and women. Overall, we find more associations between perceptions of prevalence than with the adoption of COVID-19 protective behaviors. Third, we find the fewest statistically significant relationships between HIV/AIDS outcomes and the likelihood of COVID infection, and most of the very few associations are marginally significant ($p < 0.10$). We only find a few associations between family mortality and likelihood of current COVID-19 infection among men, and total number died of AIDS and current likelihood of COVID-19 infection among

**Table 9. Multiple regression results for the association between own HIV infected status in 2006 or 2008 and COVID-19 related perceptions and responses in 2020, MLSFH data.**

| | linear | linear | linear | linear | log-linear | log-linear |
|---|---|---|---|---|---|---|
| Regression type | (1) | (2) | (3) | (4) | (5) | (6) |
| | *Dependent Variables* | | | | | |
| | Perception of the prevalence of COVID-19 in community (currently) | Perception of the prevalence of COVID-19 in community (future) | Likelihood of current COVID-19 infection | Likelihood of future COVID-19 infection | ln(Total no. of preventive actions for COVID-19) | ln(Total no. of effective preventive actions for COVID-19) |
| **HIV/AIDS in 2006/2008** | | | | | | |
| | All Adults (N = 1394) | | | | | |
| HIV infected | -0.085 | -0.182 | -0.022 | 0.094 | 0.064 | 0.063 |
| | Men (N = 606) | | | | | |
| HIV infected | -0.345 | -0.551 | -0.241 | -0.054 | 0.007 | 0.020 |
| | Women (N = 788) | | | | | |
| HIV infected | 0.006 | -0.040 | 0.070 | 0.202 | 0.067 | 0.059 |

Notes: regression models adjust for sex (in models for all adults), age, number of living children, number of household members, region of residence, level of education, and wealth quintile. The HIV/AIDS exposure measure is statistically significant at

*** $p < 0.01$

** $p < 0.05$

* $p < 0.10$.

women. Finally, our results show that actual HIV status does not predict any COVID-19-related outcomes.

Our results differ by gender: HIV/AIDS measures are more often associated with COVID-19 outcomes for women than for men. The one exception to this is family HIV mortality, where there are more statistically significant relationships among men. This result is consistent with previous research conducted during the HIV/AIDS epidemic, which shows that women actively strategized to avoid infection, often more so than men [30, 37].

Applying these results to our conceptual framework, in terms of the pathways between HIV/AIDS and COVID-19, we found the strongest support for the "mortality impact" hypothesis: experience of HIV-related mortality in the past (captured through self-reported knowledge of HIV-related mortality in the past 12 months, and ever) is positively associated with perceptions of COVID-19 presence in the community as well as preventive behaviors for COVID-19. The relationship between perceived mortality due to HIV and subsequent protective behavior from another infectious disease is generally consistent with theory suggesting that perceptions of mortality decline may have impacts on demographic behaviors, like fertility [38]. We also find that perceived prevalence of HIV in 2006 or 2008 and COVID-19 in the community in 2020 are positively associated, which provides some support for the "cumulative experience" hypothesis.

In some cases, the lack of statistically significant results was also noteworthy. We found no evidence for "direct impact": actual HIV infection status is not associated with COVID-19 related perceptions and behaviors in 2020. We also find generally weak, or no, relationship between exposure to HIV/AIDS and adopting behaviors that protect from COVID-19 and likelihood of current or future COVID-19 infection. This suggests that individuals view HIV and COVID-19 to be distinct infectious diseases and recognize their epidemiology to be separate processes. This is supported by findings from the MLSFH that respondents had reasonably high rates of knowledge about transmission pathways [14]. These results may be indicative of

the possibility that individuals view the modes of transmission for the two infectious diseases as being unrelated, or that individuals who believe themselves to be infected with HIV rationalize risky behaviors and are less likely to take up preventive actions for COVID-19.

This research has several limitations. First, although we find that individuals who perceived higher exposure to HIV in the past (2006, 2008) were more likely perceive higher presence of COVID-19 infection in their community, we cannot directly identify the underlying mechanisms that lead to these associations. Similarly, although we find that female respondents who reported that they knew of community members who died of HIV were more likely to perceive a higher prevalence of COVID in their community as well as take up preventive behaviors for COVID-19 we are not able to comment on the pathway through which greater perceived prevalence translated to uptake of preventive behaviors among women. Second, this research is not designed to be representative of the population of rural Malawi, for two reasons: although the characteristics of MLSFH respondents is similar to residents of rural Malawi from Demographic and Health Surveys data [26], the MLSFH sample was drawn only in parts of three districts within Malawi. In addition, although phone surveys offer some notable benefits over face-to-face surveys, individuals who own phones can be systematically different from those who do not in settings like rural Malawi (yet, phone surveys were the only available option to collect data during 2020 where IRB and other restrictions prevented face-to-face data collections). Third, although the longitudinal panel data are an essential asset to this research, we note that there is a substantial duration of time between the HIV/AIDS exposure (measured in 2006 and 2008) and COVID-19-related outcomes (measured in 2020), and some HIV/AIDS-related characteristics may have changed for respondents between this time.

Overall, our findings suggest that there are indeed connections between perceptions of and the behavioral response to infectious disease outbreaks: HIV/AIDS-related characteristics are associated with some COVID-19 outcomes among rural Malawians. But this applies only to selected disease characteristics, most notably the experience of infectious disease mortality, where individuals who were exposed to more mortality due to HIV/AIDS were systematically different in their COVID-19 perceptions and behaviors. There is also evidence that rural Malawians are well-informed about basic infectious disease epidemiology and see HIV/AIDS and COVID-19 as relatively distinct- which suggests that although the perceptions and behaviors related to the two pandemics are unrelated in this analysis, epidemiological knowledge gained from the HIV/AIDS epidemic may be applied to COVID-19, a connection that we do not measure here but should be the topic of future research. As the number and type of infectious diseases increases worldwide, the influence of previous infectious disease on future behaviors becomes increasingly important- and this research represents a step forward in understanding this connection.

## Supporting information

**S1 Data.**
(DTA)

**S1 File.**
(PDF)

## Author Contributions

**Conceptualization:** Philip Anglewicz, Hans-Peter Kohler.

**Data curation:** Philip Anglewicz, Iliana Kohler, James Mwera, Andrew Zulu.

**Formal analysis:** Philip Anglewicz, Sneha Lamba.

**Writing – original draft:** Philip Anglewicz, Sneha Lamba.

**Writing – review & editing:** Philip Anglewicz, Sneha Lamba, Iliana Kohler, James Mwera, Andrew Zulu, Hans-Peter Kohler.

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
