## [Decision Letter · Decision Letter 0]

25 Apr 2023

PONE-D-23-07190Is experience of the HIV/AIDS epidemic associated with responses to COVID-19? Evidence from the Rural MalawiPLOS ONE

Dear Dr. Anglewicz,

Thank you for submitting your manuscript to PLOS ONE. After careful consideration, we feel that it has merit but does not fully meet PLOS ONE’s publication criteria as it currently stands. Therefore, we invite you to submit a revised version of the manuscript that addresses the points raised during the review process.

Adhere to reviewers comment to improve your manuscript. Again follow the journal style of manuscript preparation as well as proofreading the manuscript

We look forward to receiving your revised manuscript.

Kind regards,

Ephraim Kumi Senkyire

Academic Editor

PLOS ONE

Journal Requirements:

3. We note that you have stated that you will provide repository information for your data at acceptance. Should your manuscript be accepted for publication, we will hold it until you provide the relevant accession numbers or DOIs necessary to access your data. If you wish to make changes to your Data Availability statement, please describe these changes in your cover letter and we will update your Data Availability statement to reflect the information you provide

Additional Editor Comments (if provided):

Reviewers' comments:

Reviewer's Responses to Questions

**Comments to the Author**

1. Is the manuscript technically sound, and do the data support the conclusions?

Reviewer #1: Yes

Reviewer #2: Partly

2. Has the statistical analysis been performed appropriately and rigorously? 

Reviewer #1: Yes

Reviewer #2: N/A

3. Have the authors made all data underlying the findings in their manuscript fully available?

Reviewer #1: Yes

Reviewer #2: Yes

4. Is the manuscript presented in an intelligible fashion and written in standard English?

Reviewer #1: Yes

Reviewer #2: Yes

5. Review Comments to the Author

Reviewer #1: Abstract: The abstract would have been sufficient if presented in few sentences per section based on IMRAD postulations. It will be the ideal to used Introduction, Methodology, Results and Discussion techniques, rather than a mixed-up presentation as did by the authors.

Background/Introduction

Paragraph 1: The sentence topic sound like a mixed-up not tallying with the entire paragraph. The information presented in the paragraph lacks clarity, if compared with the sentence topic.

Methodology

Malawi Longitudinal Study of Families and Health (MLSFH) should be explain using simple terms for ease of understandings and reflection for a repeat study using similar approach.

Reviewer #2: MANUSCRIPT NUMBER: PONE-D-23-07190

MANUSCRIPT TITLE: Is Experience of the HIV/AIDS Epidemic Associated with Responses to COVId-19? Evidence from Rural Malawi

Authors need to mention the ethical review of this study and if possibly attached the ethical letter acquired from the ethical board.

The study Lack figures to show relationships and correlations.

Structured questions regarding the interview and other form of data collection strategy must be submitted or attached.

Authors need to discuss about the Data analysis in the methodology. For example, the use of descriptive statistics, ANOVA etc and the software use.

In the theoretical framework, authors can also highlighted the differences between COVID-19 pandemic and HIV/AIDS epidemic in Malawi using statistical numbers like number of cases recorded, mortality, recovery etc. using secondary data is highly recommended.

Authors can list the contribution of the study at the end of the introductory section.

The abstract section can be modify by providing simple introduction to the disease before laying out the research gap/challenges.

This sentence in the abstract should be modify in this manner: Based on the outcome of the data analysis using …. Statistical approach, we found out that the experience of HIV…’

Number of COVID-19 cases should be updated as well as recovery and deaths.

The background of the study should be merged with the introduction part.

6. PLOS authors have the option to publish the peer review history of their article (what does this mean?). If published, this will include your full peer review and any attached files.

Reviewer #1: **Yes: **Umar Ibrahim

Reviewer #2: **Yes: **ABDULLAHI UMAR IBRAHIM

---

## [Author Response · Author response to Decision Letter 0]

18 Jun 2023

We appreciate the reviewers’ comments, as they have helped to strengthen this research. Below we list our responses to each comment. 

Reviewer #1

Abstract: The abstract would have been sufficient if presented in few sentences per section based on IMRAD postulations. It will be the ideal to used Introduction, Methodology, Results and Discussion techniques, rather than a mixed-up presentation as did by the authors.

We appreciate the suggestion and have reformatted our abstract into the following sections: Introduction, Methods, Results, and Conclusions. 

Background/Introduction: Paragraph 1: The sentence topic sound like a mixed-up not tallying with the entire paragraph. The information presented in the paragraph lacks clarity, if compared with the sentence topic.

We agree and have edited this paragraph to clarify our point. Essentially, we are saying that sub-Saharan Africa has a lower COVID-19 disease burden than other regions- this is now our topic sentence.

Methodology: Malawi Longitudinal Study of Families and Health (MLSFH) should be explain using simple terms for ease of understandings and reflection for a repeat study using similar approach.

We have edited the first paragraph of the Methods section to describe the MLSFH more clearly. 

Reviewer #2

Authors need to mention the ethical review of this study and if possibly attached the ethical letter acquired from the ethical board.

This study was a secondary data analysis of de-identified data. As a result, ethical approval was not required for this study. However, the broader MLSFH study did receive ethical approval to collect the data used in this analysis and the authors signed a data use agreement prior to conducting this study. We have added a description of the MLSFH ethical approval to the manuscript in the “Data” section. 

The study lacks figures to show relationships and correlations.

We appreciate this suggestion. However, given the amount of information shown in this manuscript, we feel that the most effective way to present our results is in a table format instead of using figures. But we are open to suggestions for figures to use if this reviewer has something in mind. 

Structured questions regarding the interview and other form of data collection strategy must be submitted or attached. 

We have added a link to the MLSFH survey instruments from 2006 and 2008 (available online at https://web.sas.upenn.edu/malawiresearch/documentation/questionnaires/). We have also added the MLSFH COVID-19 survey as appendix materials for this research. We note this in the “Measures” section of this manuscript. We also provide a reference for a manuscript that provides extensive detail about the MLSFH data collection strategy (Kohler et al. 2015). 

Authors need to discuss about the Data analysis in the methodology. For example, the use of descriptive statistics, ANOVA etc and the software use.

We added a description of our data analysis to the “Analytic Methods” section. This describes our analytic approach and also includes a description of the software we used for analysis (Stata version 15). 

In the theoretical framework, authors can also highlighted the differences between COVID-19 pandemic and HIV/AIDS epidemic in Malawi using statistical numbers like number of cases recorded, mortality, recovery etc. using secondary data is highly recommended.

We appreciate this suggestion and agree that it’s important to note the differences between COVID-19 and HIV/AIDS. We have added this to the paragraph to the theoretical framework section explaining why there may not be any connection between COVID-19 and HIV/AIDS, due to the differences in prevalence and mortality. 

Authors can list the contribution of the study at the end of the introductory section.

We appreciate this suggestion and have added the author contributions to this manuscript. According to formatting guidelines for PLoS One, these appear just before the references section of the manuscript. 

The abstract section can be modify by providing simple introduction to the disease before laying out the research gap/challenges. 

We have added a sentence to the abstract to introduce COVID-19, and have also added more information about COVID-19 to the theoretical framework section, to distinguish between COVID-19 and HIV/AIDS. 

This sentence in the abstract should be modify in this manner: Based on the outcome of the data analysis using …. Statistical approach, we found out that the experience of HIV…’

We appreciate this comment and have rephrased this sentence as “Based on the outcome of the data analysis using multivariable regression models, we found that the experience of HIV-related mortality…”

Number of COVID-19 cases should be updated as well as recovery and deaths.

We have updated the number of COVID-19 cases and deaths in Malawi according to 2023 data (through May). 

The background of the study should be merged with the introduction part.

We have merged the background and introduction sections, with the research goals coming at the end of this section, as is the typical format.

---

## [Decision Letter · Decision Letter 1]

22 Aug 2023

PONE-D-23-07190R1Is experience of the HIV/AIDS epidemic associated with responses to COVID-19? Evidence from the Rural MalawiPLOS ONE

Dear Dr. Anglewicz,

Thank you for submitting your manuscript to PLOS ONE. After careful consideration, we feel that it has merit but does not fully meet PLOS ONE’s publication criteria as it currently stands. Therefore, we invite you to submit a revised version of the manuscript that addresses the points raised during the review process.

In addition to reviewer's comment, please provide point by point response indicating in the manuscript where changes has occured for comments from both reviewers. 

We look forward to receiving your revised manuscript.

Kind regards,

Ephraim Kumi Senkyire

Academic Editor

PLOS ONE

Additional Editor Comments:

Dear Authors

In addition to reviewer's comment, please provide point by point response indicating in the manuscript where changes has occured.

Reviewers' comments:

Reviewer's Responses to Questions

**Comments to the Author**

1. If the authors have adequately addressed your comments raised in a previous round of review and you feel that this manuscript is now acceptable for publication, you may indicate that here to bypass the “Comments to the Author” section, enter your conflict of interest statement in the “Confidential to Editor” section, and submit your "Accept" recommendation.

Reviewer #2: All comments have been addressed

Reviewer #3: All comments have been addressed

2. Is the manuscript technically sound, and do the data support the conclusions?

Reviewer #2: Yes

Reviewer #3: Yes

3. Has the statistical analysis been performed appropriately and rigorously? 

Reviewer #2: Yes

Reviewer #3: Yes

4. Have the authors made all data underlying the findings in their manuscript fully available?

Reviewer #2: Yes

Reviewer #3: Yes

5. Is the manuscript presented in an intelligible fashion and written in standard English?

Reviewer #2: Yes

Reviewer #3: Yes

6. Review Comments to the Author

Reviewer #2: Accepted

Authors have addressed all the comments provided in the previous review and thus the article can be accepted.

Reviewer #3: All comments have been addressed. The authors only need to do thorough editing of the manuscript for grammar and typo errors.

7. PLOS authors have the option to publish the peer review history of their article (what does this mean?). If published, this will include your full peer review and any attached files.

Reviewer #2: **Yes: **Assistant Professor Abdullahi Umar IBRAHIM, PhD.

Reviewer #3: **Yes: **Dr Grace Danda

---

## [Author Response · Author response to Decision Letter 1]

25 Aug 2023

We appreciate the final comments from the reviewer and have addressed in our updated version of the manuscript. Specifically, the reviewer noted that: 

“All comments have been addressed. The authors only need to do thorough editing of the manuscript for grammar and typo errors.”

We have thoroughly reviewed the manuscript to correct grammatical errors and typos, so the final version is clear of any such issues.

We have also included our responses to earlier comments from both reviewers below, as a record of this previous effort.

---

## [Editor Report · Decision Letter 2]

14 Sep 2023

PONE-D-23-07190R2Is experience of the HIV/AIDS epidemic associated with responses to COVID-19? Evidence from the Rural MalawiPLOS ONE

Dear Dr. Anglewicz,

Thank you for submitting your manuscript to PLOS ONE. After careful consideration, we feel that it has merit but does not fully meet PLOS ONE’s publication criteria as it currently stands. Therefore, we invite you to submit a revised version of the manuscript that addresses the points raised during the review process.

We look forward to receiving your revised manuscript.

Kind regards,

Ephraim Kumi Senkyire

Academic Editor

PLOS ONE

Journal Requirements:

**Additional Editor Comments:**

Although, the authors have addressed reviewers comments successfully, I urged authors to highlight in the manuscript the such corrections  for cross-check before final decision can be made. 

---

## [Author Response · Author response to Decision Letter 2]

15 Sep 2023

Dear Sir or Madam,

Thank you for the additional comment, and I apologize for any miscommunication. To be clear, the updated submission contains two versions of the manuscript, as required by the journal. The first is an unmarked version that contains no track changes. The second is a tracked version that includes all changes tracked. I hope this meets the most recent request, but let me know if anything further is needed. 

Sincerely,

Philip Anglewicz

---

## [Editor Report · Decision Letter 3]

19 Sep 2023

Is experience of the HIV/AIDS epidemic associated with responses to COVID-19? Evidence from the Rural Malawi

PONE-D-23-07190R3

Dear Dr. Anglewicz,

We’re pleased to inform you that your manuscript has been judged scientifically suitable for publication and will be formally accepted for publication once it meets all outstanding technical requirements.

Kind regards,

Ephraim Kumi Senkyire

Academic Editor

PLOS ONE
---

## [Editor Report · Acceptance letter]

16 Oct 2023

PONE-D-23-07190R3 

Is experience of the HIV/AIDS epidemic associated with responses to COVID-19? Evidence from the Rural Malawi 

Dear Dr. Anglewicz:

I'm pleased to inform you that your manuscript has been deemed suitable for publication in PLOS ONE. Congratulations! Your manuscript is now with our production department. 

Kind regards, 

on behalf of

Prof Ephraim Kumi Senkyire 

Academic Editor

PLOS ONE